# Improved Topic Representations of Medical Documents to Assist COVID-19 Literature Exploration

**Yulia Otmakhova[1,2], Karin Verspoor[1], Timothy Baldwin[1], Simon Šuster[1]**
[1]The University of Melbourne
[2]IBM Research Australia
contact: `karin.verspoor@unimelb.edu.au`

## Abstract

Efficient discovery and exploration of biomedical literature has grown in importance in the context of the COVID-19 pandemic, and topic-based methods such as latent Dirichlet allocation (LDA) are a useful tool for this purpose. In this study we compare traditional topic models based on word tokens with topic models based on medical concepts, and propose several ways to improve topic coherence and specificity.

## 1 Introduction

As the Coronavirus 2019 (COVID-19) pandemic has presented unprecedented challenges to well-being and safety, the medical community has responded by rapidly conducting and publishing a vast amount of related research. This in turn has made it difficult for medical professionals and researchers to keep abreast of the latest evidence. Combined with the need to explore legacy literature on related coronaviruses such as SARS or MERS, there is a need for tools supporting efficient knowledge discovery and exploratory search that goes beyond simple text retrieval (Marchionini, 2006). In this context, representing and visualising the content of documents to allow the user to quickly identify relevant studies is becoming critically important.

Latent Dirichlet allocation (LDA: Blei et al. (2003)) is probably the most commonly used method for topic-based analysis of documents. It was applied by many systems in a recent Kaggle challenge over coronavirus literature,[1] and is used in search and exploration tools recently developed for COVID-19 research,[2] including our own COVID-SEE system[3] (Verspoor et al., 2020). However, while conventional LDA models work well

for topically-diverse document collections, they are less informative in narrow, knowledge-rich domains such as medicine, especially when the corpus consists of documents related to one broad topic, such as in the coronavirus-related literature. An ideal topic model should capture more specific, discriminating topics rather than generic topics made up of terms occurring in the majority of documents. In this paper, we compare topic models based on medical concepts with traditional models based on words, and examine the nature of the inferred topics in terms of genericness and coherence.

## 2 Related work

Topic modelling has been applied in biomedical domain to cluster documents (Zhao et al., 2014), improve document retrieval (Yu et al., 2016) and discover biological relationships (Wang et al., 2011) or similar drugs (Bisgin et al., 2012). In practice, however, LDA is most commonly used to discover salient topics in a document collection (see, for example, Wang et al. (2016)), including for topical representation of COVID-19 related literature (Le Bras et al., 2020; Verspoor et al., 2020). Several attempts have been made to improve biomedical topic models by extracting a controlled set of biomedical entities (Wang et al., 2011) or using MeSH headers (Doshi-Velez et al., 2014); our approach differs in that we do not use a set of known relevant terms but rather filter out noninformative words to allow for more unrestricted knowledge discovery. In terms of topic quality, AlSumait et al. (2009) introduced the notion of "junk" (incoherent) and "background" (generic) topics, which are uniformly distributed over words and documents, respectively. However, though their method allows to rank topics based on their usefulness and quality, the authors do not experiment with improving them in terms of their specificity and coherence.

---

[1] kaggle.com/allen-institute-for-ai/CORD-19-research-challenge  [2] discovid.ai, strategicfutures.org/TopicMaps/COVID-19/  [3] covid-see.com

## 3 Data

For our experiments we use the CORD-19 dataset, which is currently the most extensive coronavirus literature corpus publicly available (Wang et al., 2020). The dataset includes COVID-19 and coronavirus-related publications from various sources, such as PubMed Central open access corpus, research articles from a corpus maintained by the WHO, and bioRxiv and medRxiv pre-prints. For our dataset we use abstracts of the papers in the corpus, or the first two paragraphs of the full text if no abstract is available. We remove documents in languages other than English using the CLD2 library.[4] The resulting dataset consists of 103955 documents with the average length of 156 words.

## 4 Topic modelling

### 4.1 LDA model

Latent Dirichlet allocation represents each document as a mixture of topics, and each topic as a mixture of words (Blei et al., 2003). We use an asymmetric prior for the document–topic distributions, as it has been shown to improve the robustness of the model (Wallach et al., 2009) and coherence of the topics learned from abstracts of scientific articles (Syed and Spruit, 2018). Following Blei and Lafferty (2006), we filter out tokens or concepts which occur in fewer than 20 documents or more than in 50% of the dataset, and remove stopwords based on PubMed's list.[5] We trained models for 5 epochs, noting that convergence based on perplexity usually occurred by the fifth epoch. Preliminary experimentation identified that the optimal number of topics for the data set, based on the $C_v$ topic coherence measure (Röder et al., 2015), varied depending on the input representation (Fig. 1). To allow more direct comparison, we fix the number of topics at 25 for each model, as the coherence scores are close enough to each other at this point.

### 4.2 Document representation

We consider three different input representations of the text for inferring the models:

- **Word tokens:** The input text was tokenised using the NLTK Tokeniser[6].

- **Concepts:** Documents are transformed into an unordered set of Unified Medical Language System (UMLS: Lindberg et al. (1993)) concepts, through the application of MetaMap[7] with word sense disambiguation. All tokens that are not recognised as concepts by MetaMap are removed.

- **Non-generic concepts:** The UMLS concept-based representation of the texts, with more general concepts filtered out.

The choice of representation based on medical (UMLS) concepts is motivated as follows: (1) it avoids splitting multi-word concepts (such as *degenerative disease of the central nervous system*) into less meaningful units (*of*, *the*, *central*, etc.); (2) it ensures that disambiguated homonyms, such as *cats* (C0007450, mammal) and *cats* (C1825121, gene), can be assigned to different topics by the model; and (3) it maps different lexico-grammatical variations of a given term into a single concept, thus reducing noise in the data, and highlighting important keywords. For example, concept C0000731 occurs in the articles as *abdominal distension*, *abdominal distention*, *bloating*, *distended abdomens*, *swelling of abdomen*, etc., which would not be captured by typical approaches to text normalisation such as lemmatisation, stemming, or $n$-gram overlap. We use a bag of concept identifiers (such as [C4038448 C1314792 C1443924 C0042963 C0392760 C2948600...]) to train the model, and then represent each identifier in the results by the lexicalisation that occurs most frequently in the document collection, as distinct from the MetaMap "preferred term", which is often a technical description rather than its lexical form (e.g. we use *colon* instead of preferred term *Colon structure (body structure)*).

We also attempt to filter out generic, or broad, concepts. Scientific publications contain many non-specific terms, which can be part of their discourse structure (boilerplate sentences, section headings such as *Discussion*, phrases such as *in conclusion*), or be included in informative sentences but not be meaningful for the purposes of topic modelling. As adding all such words to a stop-word list would not be feasible, we filter the concepts based on their semantic type as defined in UMLS. Following ShafieiBavani et al. (2016) and Plaza et al. (2011), who used a similar approach to filter concepts for graph-based summarisation of medical documents, we exclude terms based on broad semantic types including QUANTITATIVE CONCEPT (*rate, unit*),

---

[4] pypi.org/project/pycld2/    [5] ncbi.nlm.nih.gov/books/NBK3827/table/pubmedhelp.T.stopwords/
[6] nltk.org

[7] metamap.nlm.nih.gov

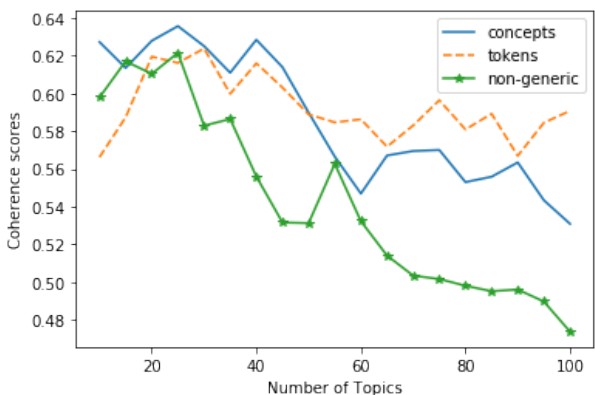

Figure 1: Coherence scores for different representations of the CORD-19 corpus.

QUALITATIVE CONCEPT (*characteristics, different*), TEMPORAL CONCEPT (*year, recent*), FUNCTIONAL CONCEPT (*use of, damage*), IDEA OR CONCEPT (*provision, health resources*), INTELLECTUAL PRODUCT (*database, review*), MENTAL PROCESS (*enable, understanding*), SPATIAL CONCEPT (*area, access*), and LANGUAGE (*italian, japanese*). We additionally exclude the following four semantic types:

CONCEPTUAL ENTITY (*example, step*), ACTIVITY (*contribute, activation*), RESEARCH ACTIVITY (*validate, research*), and OCCUPATIONAL ACTIVITY (*production, administration*).

### 4.3 Evaluation

As automatic coherence measures cannot evaluate the quality of topics in terms of how useful or representative they are, we performed human evaluation. Two annotators — one of the authors and a medical professional — judged if a topic (represented by its 5 most frequent tokens or concepts) was coherent or not; coherent topics were further subdivided into specific and generic. This distinction is important as some topics can be highly coherent, but not informative. This is especially visible in datasets where the documents are homogeneous both in terms of style (scientific articles) and content (related to coronaviruses). For example, such topics as [*research*, *study*, *approach* ] or [*coronavirus*, *virus*, *disease* ] are coherent, but not representative of the content of the paper. In line with this, each topic was assigned one of three labels by the annotators: *incoherent*, *specific*, or *generic*. Following [Newman et al. (2010)](#), to evaluate coherence, annotators were asked to decide if each topic was meaningful and interpretable. To judge specificity, they were instructed to decide if a particular set of words is

|  | Incoherent | Generic | Specific |
|---|---|---|---|
| Word tokens | 11 | 7 | 7 |
| Concepts | 3 | 6 | 16 |
| Non-generic concepts | 2 | 3 | 20 |

Table 1: Number of incoherent, generic and specific topics identified in topic models of 25 topics built over different representations of the CORD-19 corpus

likely to occur in the majority of COVID-19 related studies or not. Annotators were provided examples of incoherent, specific and generic topics related to COVID-19 but not occurring among the topics learned by the system; they did not have access to the corpus and had to make judgements regarding coherence and specificity based on their general understanding of the target issue (COVID-19). Although the definitions of the labels are somewhat vague, the Cohen's kappa for inter-annotator agreement was 0.87 (strong); disagreements were resolved by discussion. The majority of disagreements were related to deciding between generic and specific topics; though the annotators had perfect agreement for topics which can be considered generic for any medical research article, such as *article, journal, authors, published, leading*, it was more difficult to reach for topics which can be regarded as specific or generic depending on how broadly you consider the domain of interest, such as *gene, sequence, evolution, strains, coronavirus*.

## 5 Experiments

Table 1 shows the number of incoherent, generic and specific topics learned by each of the models. In general, using concepts improved topic coherence, while removing generic concepts helped to make topics even more specific.

| | |
|---|---|
| **Incoherent** | [[1], avian, [2], [3], [4]; lung, acute, or, pulmonary, blood]]; [or, other, infectious, diseases, viral]; [=, ace2, p, 95%, enzyme]; [al, dogs, canine, 2003, milk]; [[1], [3], [2], [4], water]; [r, e, o, n, d]; [will, research, information, or, many]; [t, calves, bovine, immune, mucosal]; [network, (1), (2), s, or]; [after, days, or, group, higher] |
| **Generic** | [against, vaccines, drug, activity, development]; [care, medical, health, or, healthcare]; [respiratory, syndrome, coronavirus, middle, east]; [health, public, global, infectious, world]; [analysis, methods, method, biological, sequencing]; [covid-19, sars-cov-2, coronavirus, 2019, case]; [hospital, surgery, emergency, surgical, university] |
| **Specific** | [detection, diagnostic, pcr, assay, testing]; [cell, immune, expression, viral, cellular]; [cell, antibodies, antibody, against, immune]; [protein, rna, viral, gene, sequence]; [porcine, strains, diarrhea, pedv, strain]; [risk, mortality, severe, cancer, therapy]; [respiratory, children, infections, viral, tract] |

Table 2: Baseline model topics

## 5.1 Model analysis

**Baseline model** As can be seen in Table 2, the basic (word) token-based model suffers from multiple issues, some of which can be solved by expanding the stopword list (*against*, *or*); lemmatisation (*methods/method*, *strains/strain*) or removing non-alphabetic tokens (*[1]*, *95%*), but some, such as splitting multi-word terms (*middle*, *east*; *respiratory*, *tract*) are an unavoidable result of tokenisation. Because of such splitting less specific, but more frequent parts of multi-word terms (i.e. *syndrome* in *acute respiratory syndrome*) are more likely to be generated by the model, and in the result both topics and terms in them are more generic.

**Concept-based model** The topics based on UMLS concepts are shown in Table 3. The concept-based representation helps to improve the coherence, and also to produce granular topics with more specific terms, such as *domain, peptide, residues, fusion, epitopes*. However, some issues remain, such as generic topics and non-informative terms (e.g., *associated with*) inside specific topics.

**Model based on non-generic concepts** Topics learned after filtering of broad UMLS concepts are shown in Table 4. In addition to the overall improvement in terms of specific topics, it can be noted that some of the topics generated by this model are surprisingly granular and coherent, such as AMINO ACIDS AND THEIR NAMES (*d, m, amino acid sequence, f, amino acid*), ANTI-HIV DRUGS FOR CANCER TREATMENT (*hiv, drug, development, cancer, inhibitors*), or PREGNANCY AND BIRTH (*neonatal, deliver, delivery, pregnancy, birth*).

## 5.2 Drilling into topics

Unfortunately, an LDA model cannot learn a large number of highly-specific topics, as there is a trade-off between the number of topics and their coherence. To achieve higher granularity, after we train the model, we subdivide the dataset based on the most prevalent topic in each document, and then train an LDA model on each subset. We experiment with two approaches here: the first model is the non-generic concept model as described in Section 4.1 above, while in the second non-generic concepts are re-weighted based on their log-likelihood. We treat each of the articles in the subset as a target corpus, and the remainder of its documents as a background corpus, and compare concept distributions using the log-likelihood test (Rayson and Garside, 2000). This highlights concepts that differentiate a particular document from others discussing the same broad topic, even if they have the same set of frequent terms.

After assigning log-likelihood weights to concepts in each of the documents, we sort them by weight and use the top 50% to represent the document for topic modelling, thus discarding less salient terms. Table 5 shows the top-5 topics learned by these models from a subset corresponding to topic PROTEIN BINDING (*mediated, binding, cell, pathway, receptor*). It can be seen that while the count-based topics describe general aspects of protein binding and virus replication, the topics based on the log-likelihood test refer to specific proteins and viruses. Both of these models can be useful for medical researchers, allowing them to switch between a general view of major themes in a document collection and highly-specific topics to assist drug and treatment discovery.

## 6 Conclusions and future work

In this paper, we have shown that coherence and specificity of topics can be improved through strategies that emphasise domain-targeted conceptual representations of texts. We compared word-based models to models based on medical concepts and showed that the latter, especially when only non-generic concepts are used, help to induce more informative and useful topics. We also showed that subdividing the documents by their major topic

| | |
|---|---|
| **Incoherent** | [p, 95, age, n, conclusions]; [g, m, tuberculosis, f, k]; [bronchiolitis, pregnant women, kda, pregnancy, milk] |
| **Generic** | [health, research, public health, global, challenges]; [approach, models, process, analysis, networks]; [article, journal, authors, published, leading]; [healthcare workers, recommendations, hospital, care, staff]; [countries, people, united states, estimated, hospital]; [activity, compounds, development, therapeutic, effective] |
| **Specific** | [receptor, replication, entry, interaction, ace2]; [environment, water, food, areas, transport]; [expression, activation, cytokines, immune, t cells]; [children, respiratory, rsv, respiratory viruses, associated with]; [pedv, calves, pigs, ibv, prrsv]; [mortality, associated with, severe, outcomes, ards]; [gene, sequence, evolution, strains, coronavirus]; [microbial, lung, bacteria, organisms, bacterial]; [rna, mrna, tgev, fecal, sirna]; [vaccination, antigen, antibodies, protection, challenges]; [dogs, cats, associated with, present, infections]; [infections, h5n1, bats, humans, h1n]; [domain, peptide, residues, fusion, epitopes]; [detection, assay, pcr, rapid, rt-pcr]; [antibiotic, antibiotics, cap, antimicrobial, respiratory viral infections]; [sars-cov, coronavirus, coronavirus, mers, pneumonia] |

Table 3: Topics based on concepts

| | |
|---|---|
| **Incoherent** | [form, source, center, repositories, argument]; [water, food, cattle, calves, microbial] |
| **Generic** | [healthcare workers, hospital, infection control, healthcare, staff]; [challenges, discuss, possible, support, allow]; [countries, sars-cov, world, people, public health] |
| **Specific** | [cytokines, lung, inflammation, pneumonia, treated with]; [ards, heart, lung, surgical, hypertension]; [sars-cov, coronavirus, mers, receptor, bat]; [formation, a protein, base, substrate, enzyme]; [d, m, amino acid sequence, f, amino acid]; [immune, immune response, tissues, immune system, pathogenesis]; [virion, sense, positive, rna viruses, single-stranded rna]; [applications, biological, technology, development, microbial]; [children, symptoms, fever, respiratory viruses, respiratory infections]; [hiv, drug, development, cancer, inhibitors]; [neonatal, deliver, delivery, pregnancy, birth]; [antibodies, vaccination, antigen, serum, recombinant]; [mediated, binding, cell, pathway, receptor]; [cardiovascular, cardiovascular disease, differentiation, localization, overexpression]; [birds, humans, dogs, fecal, mammalian]; [positive, detection, laboratory, sensitivity, p]; [pigs, genus, pedv, bats, virology]; [airway, lung, inhaled, aerosol, nasal]; [deaths, age, coronavirus, risk factors, hospital]; [sequence, gene, rna, genomic, dna] |

Table 4: Topics based on non-generic concepts

| | |
|---|---|
| **Counts** | [cytokines, stimulated, interferon, signaling, innate immunity]; [sars-cov, coronavirus, binding, antibodies, receptor]; [protease, cleavage, proteases, proteolytic, substrate]; [replication, viral replication, rna, positive, viral proteins]; [membrane, plasma membrane, form, mediated, binding] |
| **Log-likelihood** | [epithelial cells, ii, porcine, intestinal, sting]; [replication, h, transcription, hsv-1, phosphorylation]; [rnai, dsrna, gene expression, accumulation, polypeptide]; [n protein, cholesterol, exosomes, ebv, nucleolus]; [antigen, antibodies, t cells, integrin, cd8] |

Table 5: Top 5 topics based on counts vs log-likelihood weights

and then using the resulting subsets for topic modelling helps to learn highly specific topics which highlight general aspects of the subject matter, if frequency-based representation is used, or more narrow questions, if only the concepts with the highest log-likelihood weights are included in the model. Unfortunately, the scope of this paper did not allow us to compare the results with those of neural topic models with word-based and concept-based embeddings; we leave this question for further research.

## Acknowledgements

The authors would like to thank Antonio Jimeno Yepes and David Martinez from IBM Research Australia for their feedback and support during this project. This research was conducted by the Australian Research Council Training Centre in Cognitive Computing for Medical Technologies (project number ICI70200030) and funded by the Australian Government. The first author's work was also partially supported by the Elizabeth and Vernon Puzey Scholarship.

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
