# OpenReview forum: "Improved Topic Representations of Medical Documents to Assist COVID-19 Literature Exploration"
_EMNLP/2020/Workshop/NLP-COVID — NLP-COVID19-EMNLP Oral_

### Official Review · AnonReviewer2 · 2020-09-09
**A simple good idea that needs better comparisons**

**Rating:** 4
**Confidence:** 4

**Review:**

This paper proposes an improvement to LDA-based topic modelling by using only important concept words. It proposes to test whether limiting the vocabulary to UMLS concepts, and a smaller set of non-generic UMLS concepts improve the topics extracted by an LDA method on the CORD-19 dataset. It evaluates the quality of these topics through manual evaluation as incoherent, generic or specific. The paper finds that non-generic UMLS concepts provides improved topics.

This is an interesting and well-written paper that I found easy to understand. While I do find it believable that non-generic UMLS concepts improve over normal LDA methods, I think this paper could do more to evaluate this compared to other methods. Furthermore, it is not clear what this paper adds to the COVID-specific text mining area. I would suggest that it could be fleshed out more and sent to a more general NLP venue. The ideas proposed in Section 4.2. seem intriguing but are too quickly explained and explored.

Comments & Concerns
- The core issue is that this paper cites multiple other extensions to LDA but doesn’t attempt to evaluate them as comparisons.
- Just a note, but why aren’t both evaluation annotators authors on the paper?
- The annotation task didn’t sound pretty vaguely defined so I’m impressed by the high agreement between annotators. That’s excellent.
- Anti-HIV treatments for cancer seems like an odd topic, especially in a coronavirus corpus
- The Data section mentions concepts and non-generic terms before they are defined, which is confusing
- I know it was likely down to the page limit, but it’s weird to have the concluding remarks in the Related Work section.
- I think a stronger case has to be made for fixing the number of topics across all methods (to 25). While it makes the methods comparable using raw counts, it isn’t clear that good methods won’t be unfairly penalised by this.

---

> ### Author Response · Authors · 2020-09-29
> **Thanks for the review!**
>
> Thank you for your thorough review, we would like to address some of your questions below:
> 1. As the limitations of the short paper did not allow for an adequate comparison with other methods, our aim was not to compare with the existing methods in terms of performance but rather show how integration of medical concepts produces more useful results for covid-related research. We are working on comparing the results with that of neural topic models and will include them in the final paper if it is accepted.
> 2. The coherence score did not vary much between 20 and 30 topics for all three models, so we believe that setting the number at 25 did not adversely affect the results of any of them. We'll explain this part in more detail if necessary.
> 3. Thanks you for your remarks regarding the paper structure, it will be fixed to improve readability.

---

### Official Review · AnonReviewer3 · 2020-09-26
**Interesting analysis, but lacks substance**

**Rating:** 5
**Confidence:** 4

**Review:**

This paper describes how an LDA topic model was improved by using concepts. The evaluation - performed by humans - shows that the more refined topic models (using medical concepts) are indeed "better", meaning here more specific and less incoherent.

The conclusion is not by itself surprising: LDA models are known to be very sensitive to pre-processing (including lemmatization and stop-word removal). This is one of the reasons why embeddings became so powerful as they are less impacted by that. In this sense, comparison with a neural topic model would have been an interesting, and helpful to guide practitioners as well as to have a data-point of the usefulness of old-and-tested LDA vs more novel models.

I found the paper an interesting experience nugget... something which we don't see so much and which practitioners might benefit. The substance however is very limited, as it basically consists of a report of a human evaluation campaign. In addition, I have doubts about the following:

 - what exactly are the documents that are provided as input to the concept-LDA? Are the words enriched with concepts, or are those documents just a sequence of concepts?
 - how is a topic represented to an evaluator? By its 5 most frequent words, or some other way?
 - so what? It is not clear to me why having more "specific" topics would be helpful in the end. Can you relate the 7 specific word-topics to some of the concept-tokens (eg, by comparing the probability distribution over words)? Do they subdivide the space further, or is it a totally different soft clustering? How will this help the final application?

---

> ### Author Response · Authors · 2020-09-29
> **Thanks for the review!**
>
> Thanks a lot for your comments, they will help us clarify some points in the paper. Please find our answers below:
>
> 1. The documents are just sequences of concepts, the words which are not UMLS concepts are removed.
> 2. The topics were represented by their top 5 words, as in the tables with examples.
> 3. That's a very interesting question, unfortunately, we have not compared the distributions directly, but we'll address this in the paper if it is accepted. From the point of view of a medical user rather than NLP, though, specific topics would help to identify important documents/subsets, i.e. it would be more informative for the user if documents were marked with such topics as [n protein, cholesterol, exosomes, ebv, nucleolus] rather than, say [care, medical, health, or, healthcare]
> 4. Regarding the neural models - though we originally did not compare our models to them, we will include the comparison in the final paper if it is accepted.

---

### Official Review · AnonReviewer4 · 2020-09-27
**Interesting approach, paper feels a bit rushed**

**Rating:** 5
**Confidence:** 3

**Review:**

In this work the authors present different approaches to improve the topic coherence and specificity of topic representations using LDA with an application to the CORD-19 dataset. The authors do a great job in framing the problem and presenting their approach. Most of the details of their methodology are found in the presented text (document preparation, number of topics), but a few things could be expanded further as other reviewers have pointed out. The evaluation part is very brief and key details about the annotation/evaluation instructions and specifics of how the topics were separated into generic and specific is definitely needed. Did the manual reviewers had access to only the abstract/first 2 paragraphs of paper? or did they had access to the full paper to make their decision on topic relevance? the agreement between reviewers is quite excellent, so a bit more in-depth discussion about this would have been nice to have. The point of using humans instead of automatic metrics for the topic coherence evaluation is essential for this kind of work.  The experimental evaluation nicely characterizes the need of a certain 'guide' (while not being exactly guided-LDA) for the LDA models to be coherent by feeding them concepts of higher relevance in the clinical context. Table 1 is an excellent result. The topic tables on page 4 are nice, but should have been reduced as this didn't leave much space to the authors for a longer discussion and conclusion sections This last section seems to be missing and replaced with an odd "Related Work" section at the end which is not really a satisfactory conclusion to this paper as some of the test should be featured earlier on the paper.  Overall, very solid start of the paper that seems to run out of space near the end, and leaves important details out.

---

> ### Author Response · Authors · 2020-09-29
> **Thanks for the review!**
>
> Thank you for your detailed feedback! We will add the annotation/evaluation details we could not afford because of the space constraints and expand the discussion/conclusions as you suggested.
> Regarding your question about the annotators' access to the texts, they were not asked to do that, as evaluating topic relevance would be unfeasible for 100K documents they were learned from. However, it would be an interesting idea for the further research to sample the most representative documents for each topic and ask the annotators to evaluate their relevance.

---

### Official Review · AnonReviewer1 · 2020-09-29
**Non-generic concept based topic modeling makes the identified topics more specific**

**Rating:** 7
**Confidence:** 5

**Review:**

The work compares traditional topic models based on word tokens with topic models based on medical concepts and suggests several ways to improve topic coherence and specificity.

Latent Dirichlet allocation is the most commonly used method for topic-based analysis of documents. The authors have pointed out that conventional LDA models work well for topically-diverse document collections but they are less informative in narrow, knowledge-rich domains like medicine and specifically when the corpus consists of documents related to one broad topic, such as coronavirus-related literature.

The experiments have been conducted on the CORD-19 dataset. Three different models have been considered based on three different input representations of the text: word tokens using NLTK tokenizer, concepts based on the Unified Medical Language Systems concepts and non-generic concepts with more general concepts filtered out.

Human evaluation was performed by two annotators to evaluate the quality of topics by labelling them as incoherent, specific or generic. Although the definitions of these labels are quite vague, the Cohen’s Kappa measure of inter-annotator agreement was 0.87. The necessity of human evaluation has been well explained.

After the models are trained, the dataset is subdivided based on the most prevalent topic in each document and then an LDA model was trained on each subset. Two approached have been proposed for further experimentation: first one based on non-generic concepts and the second one based on non-generic concepts re-weighted based on the log-likelihood.

It has been empirically shown that non-generic concepts based topic modelling makes the identified topics as more specific and less incoherent and less generic.

---

> ### Author Response · Authors · 2020-09-29
> **Thanks for the review!**
>
> Thank you so much for taking time to write this detailed review, it is very encouraging!